# Nanosecond Laser Induced Surface Structuring of Cadmium after Ablation in Air and Propanol Ambient

**DOI:** 10.3390/ijms232112749

**Published:** 2022-10-22

**Authors:** Nisar Ali, Shazia Bashir, Samina Akbar, Muhammad Shahid Rafique, Ali Mohammad Alshehri, Narjis Begum, Tanveer Iqbal, Aneela Anwar

**Affiliations:** 1Centre for Advanced Studies in Physics, Govt. College University, Lahore 54000, Pakistan; 2Department of Physics, Govt. College University, Lahore 54000, Pakistan; 3Department of Basic Sciences and Humanities, University of Engineering and Technology Lahore, New Campus, Kalashah Kaku, Sheikhupura 39020, Pakistan; 4Department of Basic Sciences and Humanities, University of Engineering and Technology Lahore, Faisalabad Campus, Faisalabad 38000, Pakistan; 5Department of Physics, University of Engineering and Technology, Lahore 54000, Pakistan; 6Department of Physics, King Khalid University, P.O. Box 9004, Abha 61413, Saudi Arabia; 7Department of Physics, COMSATS University Islamabad, Islamabad 45550, Pakistan; 8Department of Chemical Engineering, University of Engineering and Technology Lahore, New Campus, Kalashah Kaku, Sheikhupura 39020, Pakistan

**Keywords:** materials, surface plasmons, phase shift, absorption, sensors

## Abstract

In the present study KrF Excimer laser has been employed to irradiate the Cadmium (Cd) targets for various number of laser pulses of 500, 1000, 1500 and 2000, at constant fluence of 3.6 J cm^−2^. Scanning Electron Microscopy (SEM) analysis was utilized to reveal the formation of laser induced nano/micro structures on the irradiated target (Cd) surfaces. SEM results show the generation of cavities, cracks, micro/nano wires/rods, wrinkles along with re-deposited particles during irradiation in air, whereas subsurface boiling, pores, cavities and Laser Induced Periodic Surface Structures (LIPSS) on the inner walls of cavities are revealed at the central ablated area after irradiation in propanol. The ablated volume and depth of ablated region on irradiated Cd targets are evaluated for various number of pulses and is higher in air as compared to propanol ambient. Fast Fourier Transform Infrared spectroscopy (FTIR), Energy Dispersive X-ray Spectroscopy (EDS) and X-ray Diffraction (XRD) analyses show the presence of oxides and hydro-oxides of Cd after irradiation in propanol, whereas the existence of oxides is observed after irradiation in air ambient. Nano-hardness tester was used to investigate mechanical modifications of ablated Cd. It reveals an increase in hardness after irradiation which is more pronounced in propanol as compared to air.

## 1. Introduction

Nanosecond pulsed laser ablation (PLA) is used for a wide range of technical and industrial applications such as laser-induced breakdown spectroscopy (LIBS) [1], laser-induced inductively coupled plasma mass spectrometry (LA ICPMS), etc. [2], material processing such as cutting, welding, drilling and surface structuring [3]. Over the last few decades, laser-induced ablation in different environments has attracted the interest of various researchers. The vaporization of target takes place during laser material interactions, which causes the generation of plasma plume. Plasma generation and evolution during laser material interactions depend on various laser parameters such as pulse duration, wavelength, nature of target, laser target distance, ambient conditions, etc. [4]. Pulsed laser ablation can be utilized for generation of structures on the irradiated targets. These nano/micro-structured materials are highly beneficial in fluidic, mechanical, electronic and optical devices due to their unique enhanced wear resistance, electrical, optical, super-hydrophobic [3] and hydrophilic [5] properties.

PLA in different environments (air and propanol in present case) is the most affective technique for the generation of surface structures as it does not produce any kind of toxic byproduct. Liquid-assisted nanosecond laser ablation of material is more preferable for the fabrication of special morphologies, nano-materials, phases and microstructures. It is also used for the preparation of single-step functionalized phases and nano/micro structures that are useful in innovative applications in display, optics, and detection fields [6] due to enhanced chemical reactivity, confinement effects and growth of small structures as compared to air. Pulsed Laser ablation in liquid (PLAL) is feasible economically for the fabrication of variety of processed materials in manufacturing industry [7,8].

In the present era, efforts are being made for the synthesis of metallic oxides and hydroxides like nickel (II) hydroxide, magnesium hydroxide, and copper (II) hydroxide, as they are used as precursors for the generation of metal oxides. Metallic hydroxides, i.e., Cadmium Hydroxide (Cd(OH)_2_), is one of the n-type semiconductors having extensive applications in the field of solar cells, sensors, batteries, super-capacitors, photodiodes, transparent electrodes and phototransistors, etc. [9]. Furthermore, Cd(OH)_2_ can also be used as a precursor for generation of cadmium selenide and Cadmium Oxide (CdO) [9].

CdO is an important n-type semiconductor with direct and indirect band gap value of 2.5 eV and 1.98 eV [10] which has outstanding applications in the fields of photo transistors, gas sensors, photocells, catalysts and transparent electrodes [11,12,13]. Due to enhanced optical properties, CdO nano/microstructures like pores and nanoparticles play an important role in biomedical applications [10,14]. Various research groups have synthesized different kinds of semiconducting materials (like CdO in present work) by using metal targets (like Cd) after irradiation in reactive ambient [15,16]. Different methods like wet chemical and chemical precipitation are employed for the fabrication of these kinds of structures. In reduction step, reducing agents are required for both methods, in turn effecting the purity of final product [17] and reducing the suitability of these nano-materials for biomedical applications [18,19]. Liquid-assisted ablation overcomes this issue and causes the generation of micro/nanostructured surfaces that are free from toxic byproducts and chemicals [20,21].

The aim of present work is to investigate the effect of laser irradiation on surface morphology, structural and mechanical properties of Cd irradiated by KrF Excimer laser. The present work provides us with a comparison of the ablation of Cd in air and alcohol (propanol) environments. To the best of our knowledge, laser-induced structuring of Cd in air and propanol environments has not been reported earlier by any research group discussing surface structuring and enhancement of mechanical properties in correlation with the changes produced in the crystallinity due to enhancement of chemical reactivity of Cd. In the present work, we study the significant impact of dry and wet environments on the surface, structural and mechanical modifications of the material. The wet ablation is always preferred over dry ablation due to confinement effects and enhanced ablation efficiency. The novelty of the present work is that it deals with the growth of various structures and enhancement of nano-hardness in correlation with the changes produced in the crystallinity due to the enhancement of the chemical reactivity of Cd. It provides us with a comprehensive insight into the physical processes and mechanisms responsible for defining ablation threshold, surface structuring and enhancement in mechanical properties after ablation in air and propanol. The present study also emphasizes the way in which Cd ablation in propanol can modify its chemical composition in more pronounced way as compared to Cd ablation in air. The second aim of this paper is to fabricate carbonates (CdCO_3_), oxides (CdO) and hydro-oxides (Cd(OH)_2_) of Cd, which makes it more useful in industrial and biomedical applications after enhancement in its strength, field emission properties and optical absorption. We have observed that propanol-assisted ablation is more suitable for the formation of carbonates of Cd (e.g., CdCO_3_). The mechanical strength of carbonate materials (CdCO_3_) is significantly higher as compared to simple metallic materials (Cd), making it more useful for industrial applications. After employment of pulsed laser ablation in liquids, the synthesis of micro/nanostructured materials becomes possible. Such materials are free from toxic chemicals and byproducts, making them useful in biomedical applications.

In the present work, SEM analysis is used to explore the surface morphology of laser ablated Cd. XRD is used to analyze the crystalline phases present on the pristine and irradiated targets, along with the measurement of the variation of crystallite Size (cs) and residual surface stresses on Cd targets after ablation in liquids. The information related to functional groups, chemical bonds, and molecular structures of atoms is obtained using FTIR analysis. The variation in chemical composition is also confirmed using EDS analysis, supporting the results of FTIR. Nano-hardness analysis is conducted for the measurement of surface hardness.

## 2. Experimental Setup

KrF Excimer laser (EX 200/125-157 GAM Laser, Orlando, FL, USA) with wavelength of 248 nm, repetition rate of 20 Hz, duration of pulse 20 nano-sec and laser pulse energy of 70 mJ was utilized for irradiation of Cd targets. The irradiated laser beam was focused through a lens of 50 cm focal length. Circular samples of Cd with thickness of 3 mm and diameter of 10 mm were used as a target material. Before irradiation the Cd targets were grinded with SiC papers with different grit numbers and polished using diamond paste to decrease the surface roughness of pristine Cd. These polished samples were then positioned into a quartz cuvette with dimensions of 10 × 10 × 45 mm^3^. Laser ablation of Cd targets was performed 5 cm before the focusing point at constant fluence of 3.6 J cm^−2^ as shown in Figure 1. Two set of experiments were performed for varying number of pulses 500, 1000, 1500 and 2000 in different environments (air, propanol). Before irradiation of Cd targets in liquid ambient, every time the cuvete was refilled with fresh liquid (propanol) with 4 mm thickness over the target surface.

The Energy of the 248 nm laser wavelength is absorbed about 6% during irradiation in propanol environment. Absorption co-efficient of propanol was calculated by using Beer-Lambert Law, i.e., $I\left(x\right)=\;I_{o}e^{-\alpha x\;}$, where the incident laser beam intensity is represented by Io (W/m^2^), absorption co-efficient is represented by α (m^−1^) and liquid thickness over the sample is represented by x. The value of the absorption co-efficient for propanol was calculated and amounts to 1.5 × 10^−2^/mm for propanol.

The laser ablation of Cd targets was carried out at constant fluence (φ) value and was calculated by using the following formula [22]:(1)$$\mathrm{Fluence}\;(\Phi )=\frac{Energy}{Area}=\frac{E}{A}.$$

Laser pulse energy was measured in mJ and area (*A*) of focused beam was measured in cm^2^ that was about 0.019 cm^2^ for the present experiment [22].

The irradiated targets were characterized by using different tools. SEM (SEM-JEOL JSM-6480 LV) analysis was used for the measurement of surface morphology, while EDS (EDS S3700N) analysis was utilized for measurement of variation in chemical composition after irradiation in both environments, i.e., air and propanol. XRD (X’Pert PRO (MPD)) was utilized for phase identification, strain/stress and dislocation density variations. FTIR analysis was carried out for identification of newly developed chemical bonds after ablation, and nano-hardness was measured by using nano-hardness tester (CSM instruments, NHT + MHT).

## 3. Results and Discussions

### 3.1. Analytical Evaluation of Laser Fluence along with Thermal Diffusivity and Ablation Threshold

The value of thermal diffusivity of irradiated target is evaluated by using the following Equation (2) [22]:(2)$$\mathrm{Thermal}\;\mathrm{diffusivity}\;(𝑎)=\frac{K}{C\rho }.$$

As Cd is very reactive in air ambient and oxidizes immediately as we place it in air, it is also confirmed from EDS analysis that about 3.34 wt% of atomic oxygen (O) is present on the untreated target. In addition, it oxidizes more even after few pulses for both ambient conditions. For this reason, we took the values of α (16 × 10^5^/cm) [23] and 𝑎 (2.925 cm^2^/s) for CdO. The value of 𝑎 is calculated in Equation (2), where ρ represents the target surface density (for CdO, 8.15 $\frac{g}{{\mathrm{cm}}^{3}}$), *C* is specific heat (0.3397 $\frac{J}{\mathrm{gK}}$ for CdO target) and K is thermal conductivity (8.1 $\frac{W}{\mathrm{mK}}$ for CdO). The value of 𝑎 comes out to be 2.95 cm^2^ s^−1^. The Ablation threshold of irradiated CdO surface is calculated by the following formulae [22]:(3)$$\mathrm{Ablation}\;\mathrm{threshold}\;(F_{\mathrm{th}})=\rho L_{v}\sqrt{t_{p}ɑ}\;,$$
where specific heat of $\frac{\mathrm{vaporization}}{\mathrm{mass}}$ is represented by $L_{v}$ with value of about 885.77 $\frac{J}{g}$ for Cd target, $t_{p}$ being the pulse duration of KrF Excimer laser of 20 ns.

Using all the values in Equation (3), ablation threshold of Cd target has been calculated to be about 1.746 J cm^−2^.

### 3.2. SEM Analysis

SEM analysis is utilized for the investigation of variation in surface morphology, average ablation depth (δ) and ablated diameter (D) of ablated region for varying number of pulses, i.e., 500, 1000, 1500, and 2000. ImageJ software is used for the evaluation of δ and D, whereas the ablated volume (*V*) is calculated by using the following formula [24]:(4)$$V=\left(\frac{\pi }{12}\right)D^{2}\delta$$

The values of variation in ablation depth, ablated diameter and volume of Cd in air for varying number of laser pulses (500, 1000, 1500 and 2000) are listed in Table 1.

From Table 1 we can see that the values of δ, D and *V* are first increased with the increase in number of pulses to 1000, while the values decreased with the increase in number of pulses to 1500, and increased again with increase in pulses to 2000. This variation in values of δ, D and *V* is related to the variation in energy absorption with the variation in number of pulses. The increase in ablated volume and depth might be related to the fact that less or no thermal conduction and less plasma expansion is taking place, causing the target surface to absorb maximum part of energy, which in turn causes an increase in values of δ, D and *V* [24]. With the increase in number of pulses to 1500, the conduction of heat on the irradiated Cd surface takes place, causing the reduction in energy absorption for target ablation. A part of incoming laser energy is absorbed by plasma during its expansion in front of target surface and this additional energy reservoir causes the reduction in ablation efficiency [25]. Pronko et al. [26] attributed this reduction in ablation efficiency to absorption of laser energy by the plasma plume which causes less energy deposition to the target surface.

Figure 2a–d show the SEM topographical images (a) untreated and at the central ablated areas of Cd for (b) 500, (c) 1000, (d) 1500, (e) 2000 laser pulses and (f) full view of ablated spot, at constant fluence of 3.6 J cm^−2^ in air. Figure 2b shows small- and large-sized cavities, along with wrinkles on irradiated surface exposed to 500 laser pulses. Figure 2c reveals the generation of cracks and formation of wrinkles across the cracking. The length of cracks decreases whereas their width increases with the increase of pulses to 1500 as shown in Figure 2d. For the maximum number of pulses (2000), cracking width and wrinkles become more distinct and increase significantly. This can be related to heat accumulation effects caused by enhanced laser energy absorption and incubation effects with the increase in number of pulses. During laser irradiation, material heats causing the enhancement in recoil pressure, and material flows outwards causing generation of wrinkles [27]. The presence of surface wrinkles is related to the presence of compressive stresses on the target surface (confirmed from XRD due to higher angle shifting as compared to stress free data), whereas surface cracking and presence of cavities can be related to the presence of tensile stresses (also confirmed from XRD analysis due to lower angle shifting as compared to stress free data) [28].

Surface cracking in Figure 2c–e can also be related to the presence of thermal stresses [29]. Multiple-pulse-laser irradiation causes accretion in thermal stresses and in turn weakens the atomic bonds and generates cracks and cavities on the irradiated target surface [29]. For Cd exposed to maximum number of pulses of 2000, pronounced cracking due to increase in number of pulses is observed, while Figure 2f shows the full view of ablated spot for maximum number of pulses, and inset of figure shows the magnified view from boundary of ablated zone. 

Figure 3a–d show the SEM images at periphery of laser-ablated areas of Cd targets exposed to (a) 500, (b) 1000, (c) 1500 and (d) 2000 laser pulses, at constant fluence value of 3.6 J cm^−2^ in air. Variation in number of pulses strongly affects the surface structuring. For minimum number of pulses (500 pulses), surface clusters with islands and flake-formation are observed as shown in Figure 3a. With the increased number of pulses of 1000 (Figure 3b) the surface clusters are converted into nano/micro particles. For a smaller number of pulses (500 pulses), the plasma plume obtains low plasma pressure, temperature and a lesser amount of homogeneous density distribution inside plasma [30]. Therefore, this weak plasma is strongly influenced by ambient environment (air); in turn, low density nano/micro flakes, surface clusters and nano/micro-particles are generated [31]. During nanosecond laser ablation, the ablation plume presents conditions that are very suitable for the generation of clusters. Enhanced ionization leads to extraordinary enhanced nucleation and smaller critical radius which can cause the generation of surface clusters [32,33]. The size of structures formed on the irradiated target surface after laser irradiation depends basically on two mechanisms, i.e., explosive ejection and thermal evaporation [34]. 

The size of nano/micro particles also depends on concentration and temperature of ablated species when the vaporization mechanism prevailed at low radiation intensity (or lesser number of pulses). A formation rate and nano/micro particle’s size is determined by the temperature and concentration of the ablated substance [34,35].

Due to explosive ejection, larger-sized particles (10’s of nm) are generated. If vaporization phenomenon becomes dominant, then more dispersed particles (several nm in size) are generated [34].

Generally, the ablation mechanism can be divided into three steps. During laser irradiation, the target material gets heated at temperatures higher than its critical temperature value and vaporization takes place. In second step, the evaporated material generates a thin and dense plasma plume containing ions, electrons and neutral species. The laser-generated plume absorbs laser energy via inverse Bremsstrahlung and photo ionization and causes the growth of pressure in the plume. This pressure growth accelerates the plume to enhanced velocities perpendicular to the irradiated target surface. In the last stage (step), the laser-induced plasma expands adiabatically after the termination of laser pulse [36,37].

The plasma dynamics are affected by re-emitted plasma energy that is coupled back to the target surface [38], but no explosive boiling takes place. Kelly et al. [5] reported in their work that phase explosion takes place when the plasma temperature becomes more than 0.9 T_c_ (the critical temperature). This kind of ablation can cause the generation of micro/nano particles in the plasma plume that can be re-deposited on the target surface in the presence of air ambient.

For 1500 pulses, the particles are converted into nanowires and nanorods, which are grown on the top of clusters and cavities as shown in Figure 3c.

The generation of nano-wires occurs when the materials are ablated in certain controlled environments (air in present case) as shown in Figure 3c. Morales et al. [39] reported in their research that ablation of laser-irradiated metallic targets can produce nano-particle seeds for the generation of nanowires. The growth mechanism of nanowires depends on laser supported vapor–liquid–solid (VLS) mechanism. According to this model, vapor phase gives rise to a seed particle for the formation of liquid catalyst that causes the generation and growth of nano-wires until they are cooled to solid states [40,41]. Surface generated plasmon–plasmon interactions during laser ablation can join nano particles and produce nano-wires [40]. The distribution and size of nano wires might be controlled by controlling the ablation criterions like laser fluence/number of pulses (in present case), size of seeding nano-particles and inlet gaseous pressure [31,40]. With the increase in number of pulses to 2000, enhancement in the surface temperature causes the increase in density of nano-wires and cavities.

The values of variation in ablation depth, ablated diameter and ablated volume in propanol ambient for varying number of laser pulses (500, 1000, 1500 and 2000) are listed in Table 2.

With the increase in number of pulses the value of δ, D and *V* is first increased with the increase in number of pulses to 1500, while it decreases with the further increase in pulses up to 2000.

Comparison of Table 1 and Table 2 shows that ablation depth is smaller in case of propanol as compared to air. Contrary to the dry treatment, presence of liquid presents a superior heat sink by effectively cooling the target surface, and as a result reduces the excessive heating due to the shielding effect and causes less energy deposition and rapid cooling [42]. Figure 4a–d show the SEM topographical images at the center of laser-ablated areas of Cd targets for (a) 500, (b) 1000, (c) 1500 and (d) 2000 laser pulses, at constant fluence of 3.6 J cm^−2^ in propanol.

Figure 4a shows the surface morphology after ablation with 500 pulses, which displays the presence of nanoparticles, protrusions, cavities, pores, debris and re-solidification of material after melting. With the increase of pulses to 1000 (Figure 4b), formation of cavities, pores and channels along with debris across the cavities is observed. 

The formation of debris, their shape and size on irradiated target surface depends generally on material properties and on ablation time step. Subsurface boiling during laser irradiation and explosion [43] in the liquid ambient (due to shielding effect of liquid) may re-deposit the droplets and debris on the irradiated target surface.

Nanosecond laser ablation in liquid confined environments generates plasma with high temperature and pressure [44,45]. This can generate dense electron cloud inside the target interaction region that can transfer energy to the lattice through electron–phonon relaxations [46]. The heat redistributes via lattice vibrations and energy is steered into the Cd target. This energy conduction can melt and vaporize the irradiated target surface. This kind of vaporization in a liquid environment can trap many bubbles. The explosive boiling of these bubbles in the liquid can generate cavities [47] and some trapped bubble sites that can also appear in the form of debris and protrusions [48]. The size variation of observed protrusions is attributed to nonuniform energy absorption at different lattice sites due to voids, inhomogeneities and impurities [49].

After irradiation of the target with an increased number of pulses to 1000, circular holes/cavities and channels are generated. The reflection of laser beam from side walls to initially formed hole and presence of recoil pressure introduced by evaporated liquid in the target ambient causes the generation of hot spots on irradiated target surface, consequently resulting in channel-formation [50].

The increase of pulses to 1500 (Figure 4c) destroys the previously developed channels and shows enhancement in pore/cavity density. The subsurface overheating during laser pulse irradiation and subsequent explosion [43] causes deposition of a mixture of vapors, irregular debris and droplets. After the termination of laser pulse, the thermal diffusion and subsequent explosive boiling are potential mechanisms. Nucleation of bubbles causes the generation of cavities after laser ablation in liquid confined environments. The surface porosity and cavity density get enhanced with the increase of pulses from 500 to 1500 (Figure 4a–c). Circular cavities (Figure 4b,c) are basically the sites of bubble nucleation that are escaped from molten irradiated surfaces through subsurface boiling. During irradiation, some bubbles escape, while others cannot escape the liquid and re-solidify in the form of debris.

Figure 4d shows much planer surface after exposure of Cd to 2000 pulses, the density of pores and cavities reduce due to refilling by shock-liquefied material at maximum number of pulses in liquid ambient. 

Figure 5a–d show the SEM topographical images at boundary of laser-ablated areas of Cd targets after exposure to (a) 500, (b) 1000, (c) 1500 and (d) 2000 laser pulses, at constant fluence of 3.6 J cm^−2^ in propanol. 

Figure 5a shows the traces of subsurface boiling, channels, pores, cavities and cavities with LIPSS after irradiation to 500 laser pulses. With the increase of pulses to 1000 shown in Figure 5b, the density of cavities, channels and pores enhances, while LIPSS inside cavities disappear due to more energy absorption and surface melting. Here the ripples get destroyed after enhancement of pulses from 500 to 1000 due to more energy absorption and melting, and develop again in more distinct form with the increase in pulses to 1500.

For 1500 pulses (Figure 5c), an enhancement in the size of channels and cavities is perceived. Some traces of micro-conical structures and generation of LIPSS inside these cavities is also observed. Generation of LIPSS on a laser-ablated surface can be related to the interaction of laser radiation with the surface-scattered waves [29,51]. The presence or excitation of Surface Plasmon Polaritons (SPPs) introduces periodic improvement in the local field on surface layers. SPPs are observed due to interaction of incoming laser field with the free electron cloud generated on the material surface [52].

The presence of rounded ripples at boundary area of irradiated spot in propanol ambient can be related to the bubble formation and expansion process [29]. Bubbles can generate LIPSS using different mechanisms as revealed in Figure 5a,c. As the laser radiation diffracts from the bubbles, it may generate the rings of light intensity on irradiated metal surfaces. In addition, the heat of dissociation and vaporization essential for creation of bubbles at metal liquid interface cools locally at the irradiated target surface and excites the capillary waves on the molten Cd via Marangoni flow [53]. Dissociation and vaporization phenomenon can remove the thermal energy on the metallic molten surface just underneath the bubble and cool it abruptly. The irradiated surface tension reduces with the increase in temperature, the surrounding hot liquid drifts to cooler regions and deforms laser-irradiated surface [54], while the deformation process can generate the circular capillary waves. When the capillary waves superimpose with each other due to laser irradiation of target surface by multiple pulses, LIPSS inside the cavities may generate as shown in Figure 5a,c. The channels convert into wave-like ridges, hillocks and conical microstructures and some rounded cavities are also seen with the increase of pulses to 2000 (Figure 5d). These kind of surface structures are formed on the irradiated Cd target surface due to high energy absorption via heat deposition and partial melting on the boundary area of the ablated region after laser ablation. These extruded structures are formed mainly due to thermal expansion and elastic rebound of lattice on the melted target material that is pushed away on the target surface [55,56]. Various physical deformation mechanisms, such as material ablation, melting and evaporation, generate the conical structures [57]. All these mechanisms depend on laser fluence, thermal conductivity, diffusivity, heat capacity of target material and ambient environment [57].

### 3.3. XRD Analysis

Structural analysis of un-irradiated and laser-ablated Cd targets in air ambient is shown in Figure 6a–e. The reflection planes of Cd (JCPDS: 00-005-0674) at 2θ values of 31.83°, 34.73°, 38.33°, 47.75°, 61.05°, 62.30°, 66.59°, 71.57°, 73.24°, 75.46°, 77.26° having hkl values of (002), (100), (101), (102), (103), (110), (004), (112), (200), (201), (104) are shown in Figure 6a. After irradiation with 500 pulses, new peaks of cubic CdO (JCPDS: 05-0640) at angular positions of 38.08°, 61.94°, 66.21°, 76.96° and with hkl values of (200), (311), (222) and (400) are identified. Rhombohedral Cadmium Carbonate CdCO_3_ JPCDS (00-008-0456) at angular position 34.27° with hkl of (006), cubic CdO_2_ [58] appearing at 47.45° having hkl value (220) and a mixture of CdO + CdO_2_ with hkl values (311) + (023) at 60.81° are shown in Figure 6a.

Crystallite size (c.s) [59], dislocation density (d.d) [60], residual stresses and lattice parameter (a) [61,62,63] for CdO (200) peak appearing at 2θ value 38.08° are shown in Figure 6b–e, respectively. Young’s Modulus value used for the calculation of residual stresses for CdO is 50 GPa [64].

During laser ablation, implantation of ions from plasma on the irradiated target surface and laser generated thermal shocks produces compressive and tensile residual stresses, respectively, on the irradiated target surface. Presence of these stresses can also be confirmed from shifting of the diffraction peaks to higher and lower angular position of irradiated targets as compared to stress-free data [61]. The presence of these stresses can also be related to the production of lattice strains generated due to ion implantation during laser irradiation to the interstitial cites generating lattice defects and thermal shocks [61].

In present case, the c.s is inversely proportional to dislocation line density and is directly related to lattice parameter. This shows that enhanced value of c.s causes reduction in d.d and enhances value of lattice parameter, whereas the reduced c.s will be responsible for increasing d.d and decreasing lattice parameter [62], Figure 6b–e.

The reduction in c.s, lattice parameter and peak intensity along with an increase in dislocation line density is observed with the increase of number of pulses to 1500, and an increase is observed in case of air with the increase of pulses up to 2000. The residual stresses are relaxed from tensile to compressive with the increase of pulses from 500 to 1500, then again convert to tensile with increase of pulses to 2000 (reduction in cracking size and. comparatively, compact surface can be seen from SEM analysis (Figure 2c)).

The c.s is reduced with an increase in number of pulses due to intensified energy absorption and re-crystallization after laser irradiation. Increased number of pulses from 500 to 1500 causes intensified energy absorption and ionization of the ambient. The target surface thus enhances collisions between evaporated Cd and ionized ambient species, which in turn reduces the energy of involved species. These low energy species can diffuse into CdO lattice causing the reduction in c.s via broadening of peaks, causing lattice distortion, generating point defects. This, in turn, decreases the c.s and lattice parameter from 4.73 Ǻ to 4.38 Ǻ, along with reduction in peak intensity by inducing the generation of compressive stresses in CdO phase (200) [62,65]. With the further increase of pulses to 2000, increase in c.s of CdO (2 0 0) plane is due to enhanced energy of particles, and surface temperature rise in turn increases c.s and peak intensity through conversion of compressive stresses to tensile. The lattice defects and micro-strains once again increase the lattice parameter to 4.73 Ǻ [62]. With an increase in number of pulses, thermally-generated shock waves cause the generation of residual tensile stresses [61].

XRD diffractograms perceived from Figure 7 show the presence of oxides and hydrides of Cd for targets irradiated in propanol ambient. The reflection planes of un-irradiated Cd at 2θ values 31.67°, 34.66°, 38.25°, 47.82°, 61.155°, 62.167°, 66.43°, 71.57°, 73.28°, 75.46°, 77.38° with hkl values of (002), (100), (101), (102), (103), (110), (004), (112), (200), (201), (311) are shown in Figure 7a. After irradiation with 500 pulses, new peaks of hexagonal Cd(OH)_2_ pattern number JCPDS (01-073-0969) appear at an angle of 75.32° with hkl values of (202), cubic CdO (JCPDS: 05-0640) at angular positions 31.79°, 38.135°, 66.3° with hkl values (−111), (200) and (222), the mixture of rhombohedral CdCO_3_ JPCDS (00-008-0456) and hexagonal Cd(OH)_2_ pattern number JCPDS (01-073-0969), at angular position 34.75° with hkl value of (113) + (002), cubic CdO_2_ [58] appearing at 47.88° and 61.13° having hkl value (220) and (023) and a mixture of CdO + Cd(OH)_2_ with hkl values (311) + (101) at angular position 62.31° are identified and shown in Figure 7a.

The c.s, d.d, residual stresses and a for CdO (200) peak appearing at 38.135° are evaluated from XRD data and are shown in Figure 7b–e, respectively. With the increase of number of pulses from 500 to 1000 in propanol ambient, increase in peak intensity along with the increase in c.s and a is observed. The enhancement in peak intensity is related to the atomic diffusion in the grain boundaries and increase in c.s, or may be related to the enhancement of X-rays diffraction from the target surface [61,66] (Figure 7). With further increase in number of pulses to 2000, the reduction in peak intensity and ‘a’ is observed owing to re-crystallization phenomenon due to melting and re-solidification causing the enhancement in dislocation line density on irradiated target surface. Enhancement in tensile stresses with the increase of number of pulses from 500 to 1000 is observed and is shown in Figure 7d.

This can be related to the enhancement in cavity density and cavity size with the increase in pulses (Figure 4a,b), while further increase of pulses to 1500 and 2000 shows the relaxation of tensile stresses to compressive. This can be related to the annealing effects generated due to multi-pulse laser irradiation [61]. Increase in number of pulses basically reduces the initially generated residual stresses and defects through annealing. These results match well with SEM results showing the reduction in cavity density and comparatively planer surface is observed as shown in Figure 4d.

The comparison of dry (air) and wet (propanol) ablation of Cd reveals the formation of carbonate and oxides in air ambient. The propanol-assisted ablation is responsible for the formation of hydrides along with the generation of carbonate and oxides. In addition, for both ambient variations in c.s, peak intensity and residual stresses are observed. After irradiation in air, maximum residual tensile stresses, enhanced c.s, and generation of high density of cavities and cracking can be seen in Figure 2 and Figure 6 as compared to propanol with maximum compressive and less tensile stresses causing compact surface with low defect density and cracking on irradiated target surface (Figure 4 and Figure 7).

### 3.4. Hardness Analysis

Nano-indenter was utilized to measure surface hardness with respect to cs of nanosecond laser-ablated Cd targets after irradiation for various pulses (500, 1000, 1500 and 2000) in air ambient (shown in Figure 6b). Hardness measurement is basically utilized to determine the material’s resistance to plastic deformation. The hardness value of unexposed Cd is about 0.20 GPa. Hardness value of Cd targets after irradiation in air with 500 laser pulses is increased up to 0.4 GPa, 2 times greater than untreated value.

With the further increase of number of pulses to 1500. An increase in hardness value up to 0.65 Gpa is achieved that is 3.25 times greater than the hardness value of untreated Cd target. The increase in hardness can be related to the refinement in grain/crystallite size, causing the increase in dislocation density, hindering further dislocation motions. This, in turn, causes an increase in surface hardness [67,68]. Increase of hardness can also be explained on the basis of laser-generated shock waves generated by the expansion of elevated plasma pressure by the end of laser pulses. After laser irradiation, the vaporization of target takes place, which causes the generation of high temperature and high pressure plasma (A. M. Mostafa 2017). Plasma expansion during irradiation introduces pressure and compressive waves that cause propagation of shock waves on irradiated target surface. The material gets plastically deformed due to enhancement in peak pressure when it becomes higher than dynamic yield stress/strain. The generation of compressive stresses and material’s resistance to fatigue corrosion and cracking is enhanced [69]. In addition, the reduction in grain/crystallite size provides enhancement in density of grain boundaries with the increased number of pulses that hiders the motion of dislocations and enhancement of surface hardness is perceived (Figure 6b).

With further increase in number of pulses up to 2000, the reduction in surface hardness to 0.48 GPa is achieved with the increase of cs. This phenomenon of reduction in surface hardness can be related to the presence of residual stresses (tensile) as shown in Figure 6b,d. Tensile residual stresses cause the increase in cavity size and surface cracking (Figure 2 and Figure 3) as well as grain growth, and in turn reduces surface hardness [22].

Here we observed that the hardness of irradiated targets in air for any number of pulses is higher as compared to un-irradiated Cd targets. This can be related to the effect of diffusion of O on lattice and interstitial sites. That causes the enhancement in lattice distortions and reduction in cs, which in turn increases surface hardness [69]. Reduced hardness value for maximum number of pulses compared to previously irradiated Cd targets is due to O diffusion across boundaries of grains, which causes the enhancement in cavity density and size as shown in Figure 2 and Figure 3.

Figure 7b shows the change in hardness depending on c.s with the increase in number of pulses in propanol ambient. For 500 laser pulses, the hardness value of Cd surface is increased by 0.37 GPa, which is 1.85 times more than un-irradiated target. With the increase in number of pulses to 1000, reduction in hardness value (0.316 GPa) with the increase of c.s is observed. Presence of tensile stresses on the irradiated target surface is responsible for this increase in tensile stresses, as shown in Figure 7b,d. Further increase in number of pulses to 2000 causes the enhancement in hardness to 0.7 GPa (i.e., 0.35 times the un-irradiated hardness value) with the refinement in c.s due to the presence of compressive residual stresses. Here the increase in surface hardness is related to the diffusion of atomic oxygen (O) and carbon (C) to the lattice and interstitial sites during irradiation in the propanol ambient. This causes the reduction in cs due to increase in lattice distortion and in turn increases surface hardness. Reduction in hardness can be related to the diffusion of O/C across boundaries of grains that causes enhancement in cavity density and size, as shown in Figure 4 and Figure 5.

Comparison of two ambients (air & propanol) shows the hardness in propanol is higher as compared to air ambient. This can be related to more diffusion of atomic carbon to lattice and interstitial sites due to less atomic size as compared to oxygen (confirmed from EDS and XRD analysis), causing the enhancement in surface hardness in propanol as compared to air ambient.

### 3.5. FTIR Analysis

FTIR analysis is a very important and non-destructive technique which provides precise measurement to study the molecular structure and chemical bond between atoms. Each peak obtained by FTIR indicates a functional group. In Figure 8, the infrared spectrum of Cd is presented with a varying number of pulses in the air ambient. Each labeled peak in the graph shows a functional group. Various peaks identified at wavenumbers of 712 cm^−1^, 765 cm^−1^, 857 cm^−1^, 925 cm^−1^, 1120 cm^−1^, 1188 cm^−1^, 1390 cm^−1^ indicate the presence of CdO [70]. Adnan et al. [9] also claimed that the peaks appearing in the range of 1386 to 1400 cm^−1^ belong to CdCO_3_ that is also confirmed from XRD results.

Peaks at higher wavenumbers 1567 cm^−1^, 2107 cm^−1^ and 2337 cm^−1^ correspond to Aromatic (C=C), (-C≡C-) and Carbon Dioxide (O=C=O). With the increase of pulses to 1000, the increase in average peak intensity of previously generated bonds is observed. For further increase in pulses, reduction in average peak intensity is seen and is shown in Table 3. In addition, the peak appearing at wavenumber of 765 cm^−1^ vanishes along with the development of a new peak of Carbonyle (C=O) appearing at 1735 cm^−1^. This means that an aromatic functional group is present for any number of pulses. The change in the concentration is indicated by different absorption bands [71].

With increasing or decreasing the strength of treatment (number of pulses), the intensity of bands varies. These structural variations are due to different treatment parameters [72]. If a material is rich in some specific functional group, then the peak of that particular functional group repeats again and again. The reason behind shifting may be inter- or intra-molecular bonding and chain Vander Waals interactions [73].

The enhancement in peak intensities can be related to the adsorption of gases into the surface of Cd that are present in air, or may be related to the material (Cd) surface oxidation after irradiation in air ambient [74]. In turn, reduction in the peak intensities can be related to decrease in intermolecular bonding due to enhanced degradation or due to some other chemical variations [75].

Figure 9 shows the infrared spectrum of Cd with varying number of pulses from 500 to 2000 in propanol ambient. Each labeled peak in the graph shows a functional group. Various peaks identified at wavenumbers of 716.18 cm^−1^, 851.02 cm^−1^, 1120.4 cm^−1^, 1397.95 cm^−1^, and 1456.92 cm^−1^, indicate the presence of CdO [70]. The peaks appearing in the range of 1386 to 1400 cm^−1^ belong to CdCO_3_ formation, whereas additionally identified peaks at higher wavenumbers of 1593.3 cm^−1^, 1692 cm^−1^, 1745 cm^−1^, 1994 cm^−1^, 2121 cm^−1^ and 2331 cm^−1^, correspond to Cd-OH, C-C, Carbonyle (C=O), Hydrides of metal (Cd-H) and Carbonyle (C=O) [76,77]. The variation in the average peak intensity of IR absorption spectra is shown in Table 4.

Comparison of air and propanol ambient reveals the existence of CdO in the range of 700 to 1500 cm^−1^ under both ambient conditions. These peaks (metal-oxides) are more intense in air as compared to propanol ambient, showing that the bond dissociation energy in air is greater as compared to propanol [77]. In addition, the peaks of Carbonyle (C=O) are appearing in both ambients. A low intensity Carbonyle (C=O) peak appearing during irradiation in air ambient might be due to the reaction between impurity element carbon present in Cd or due to presence of small content of carbon, already present in air. During ablation in propanol, the high intensity peak of Carbonyle (C=O) is developed due to the presence of C in propanol ambient.

Generation of micro- and nano-scale structures shown in SEM micrographs (Figure 2, Figure 3, Figure 4 and Figure 5) are well-correlated with the formation of oxides on the target surface after irradiation in air and oxides and hydrides after ablation in propanol ambient.

During laser ablation of Cd in air and propanol ambient, the photo-chemical and photo-physical processes become dominant and metal-ambient reactions take place. Laser-induced structuring of metals through laser irradiation in liquids/solution is extensively reported and studied, while formation of hydroxides and oxides involves complex processes [78,79]. Laser ablation in liquid ambient generates hot plasma in vicinity of laser spot on target surface. During laser irradiation in gaseous ambient, plasma is confined in gasses, and during the pulse, adiabatic expansion of plume with supersonic speed takes place, creating shock waves in front of plasma plume, generating elevated pressure and further enhancing the temperature of plasma. Liquid is denser as compared to gaseous ambient, therefore enhanced pressure is experienced by expanding plume.

Consequently, the plasma temperature and pressure becomes significantly higher as compared to ablation in gaseous ambient [80]. In front of plume expansion, the enhanced pressure creates a suitable ambient for enhancing reaction between ablated species with oxygen and OH^−^ ions dissolved in liquid medium. In the present experiment, active Cd ions, clusters and atoms react with dissolved oxygen at alliance plasma regime and liquid, whereas OH^−^ ions generated via laser-induced breakdown of propanol for each laser pulse producing CdO & Cd(OH)_2_ molecules.

For air ambient:(5)$$2\mathrm{Cd}+O_{2}\to 2\mathrm{Cd}$$
(6)$$\mathrm{CdO}+{\mathrm{CO}}_{2}\to {\mathrm{CdCO}}_{3}.$$

For propanol ambient:(7)$$2{\mathrm{CH}}_{3}{\mathrm{CH}}_{2}{\mathrm{CH}}_{2}\mathrm{OH}+\mathrm{Cd}\to \mathrm{Cd}{\left(\mathrm{OH}\right)}_{2}+{\mathrm{CH}}_{3}{\mathrm{CH}}_{3}{\mathrm{CH}}_{3}{\mathrm{CH}}_{3}{\mathrm{CH}}_{3}{\mathrm{CH}}_{3}$$
(8)$$\mathrm{Cd}{\left(\mathrm{OH}\right)}_{2}\to \mathrm{CdO}+H_{2}O$$
(9)$$\mathrm{CdO}+{\mathrm{CO}}_{2}\to {\mathrm{CdCO}}_{3}.$$

Presence of these elements is also confirmed from XRD results (Figure 6 and Figure 7).

Here presence of C=O stretching vibration bond is related to the existence of enol type complexes, i.e., metal acetylacetonate compounds, due to reaction between alcohol (propanol) with Cd. Laser ablation of Cd in propanol ambient causes the enhancement in reaction rate between metallic ions and propanol molecules, which causes the formation of clusters Cd^+2^(CH_3_CH_2_CH_2_OH)_n_. Here association products are being stabilized by evaporation of one or more molecules that can be easily understood from the following formula [77]:(10)$$M^{+2}A_{m}\to M^{+2}A_{n}+A_{m-n}\;(n<m-1).$$

Here $M^{+2}$ is the metal ion such as Cd^+2^, while A represents the alcohol. The interaction between $M^{+2}$ and m alcohol molecules $A_{m}$ forms $M^{+2}A_{n}$ and $A_{m-n}$ (the number of molecules that do not take part in reaction) [77].

### 3.6. EDS Analysis

EDS analysis is utilized to study the chemical composition of unexposed and laser-irradiated Cd targets shown in Table 5 for 1000 laser pulses. Untreated Cd sample shows the following elements: Cd ~90.36 wt.%, balanced to 100% by C (~1.18 wt.%), N (~1.33 wt.%), O (~3.34 wt.%), Al (~0.96 wt.%), Si (~0.11 wt.%), W (~0.23 wt.%), and Protactinium (Pa (~2.49 wt.%)).

Complete removal of Al, Si and W is observed after irradiation with 1000 laser pulses in both air and propanol ambient. The variation in chemical composition of almost all elements is given in Table 5. The Cd content is decreased from 90.36 wt% to 76.45 wt% after irradiation in air and 75.20 wt% in propanol. The value of C content is decreased in air and increased in propanol (confirmed from FTIR analysis). Atomic nitrogen and oxygen content is increased in both ambient environments. Heating Cd with overlapping laser pulses causes more diffusion of O into the surface, therefore causing the formation of oxides on irradiated target surface (confirmed from SEM, FTIR and XRD analysis). The comparison of both ambients shows that oxidation level in air irradiation is higher than in propanol and confirmed from re-deposition in air ambient.

Every element responds differently due to different photo-physical, photo-chemical and photo-thermal properties after irradiation with laser pulses. This may cause uneven energy deposition for multi-pulse ablation which can generate hillocks, debris, cones, LIPSS, dendrites and wavelike ridges on the irradiated Cd surface (Figure 2, Figure 3, Figure 4 and Figure 5).

## 4. Conclusions

The significant effects of the nature of dry (air) and wet (propanol) environments on variations in surface morphology, crystallinity, chemical composition and nano-hardness of laser ablated Cd for a varying number of pulses (500, 1000, 1500 and 2000) at a constant fluence of 3.6 J cm^−^^2^ were evaluated in this study. The surface structuring and enhancement of mechanical properties are discussed in correlation with the changes produced in its crystallinity due to the increase in the chemical reactivity of Cd in both ambient environments. The observed variations in chemical composition are responsible for dissimilarities in surface structuring after ablation in both media. The surface features, chemical composition, structural and compositional analysis, and mechanical properties of irradiated targets are evaluated using a Scanning Electron Microscope (SEM), X-ray Diffraction (XRD), Fast Fourier Transform Infrared spectroscopic (FTIR), Energy Dispersive X-ray Spectroscopy (EDS), and a Nano-hardness tester. The surface morphology explored by SEM analysis clearly reveals some common features, e.g., cavities, pores and micro- and nanoparticles for both environments. However, air-assisted ablation is responsible for the growth of nanowires, micro rods and clusters. These structures were completely absent in propanol-assisted ablation of Cd. When Cd is ablated in propanol, LIPSS inside circular cavities and wrinkles were observed. The ablated diameters, depth and volumes of laser-irradiated Cd are higher in air than propanol. XRD and FTIR also confirm the formation of CdO, CdCO_3_ in case of air, whereas, in the case of propanol, new hydro-oxide phases (Cd(OH)_2_) are observed along with CdO, CdCO_3_. The observed variations in chemical composition are responsible for dissimilarities in surface structuring after ablation in both media. The hardness increase of Cd is more pronounced in propanol due to increased C diffusion in propanol as compared to in air (confirmed from EDS analysis). It is comprehensively concluded from these results that dry ablation is more beneficial in micromachining, drilling and cutting of materials. Wet ablation is more favorable for the increase in surface hardness and formation of carbides and hydroxides, making it beneficial in industrial and biomedical applications.

## Figures and Tables

**Figure 1 ijms-23-12749-f001:**
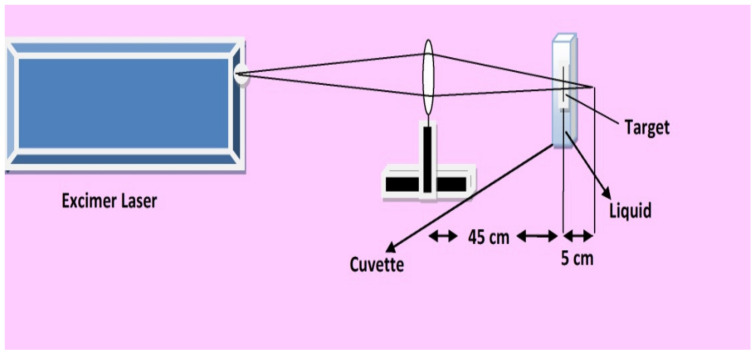
The schematic diagram of the experimental setup for laser ablation.

**Figure 2 ijms-23-12749-f002:**
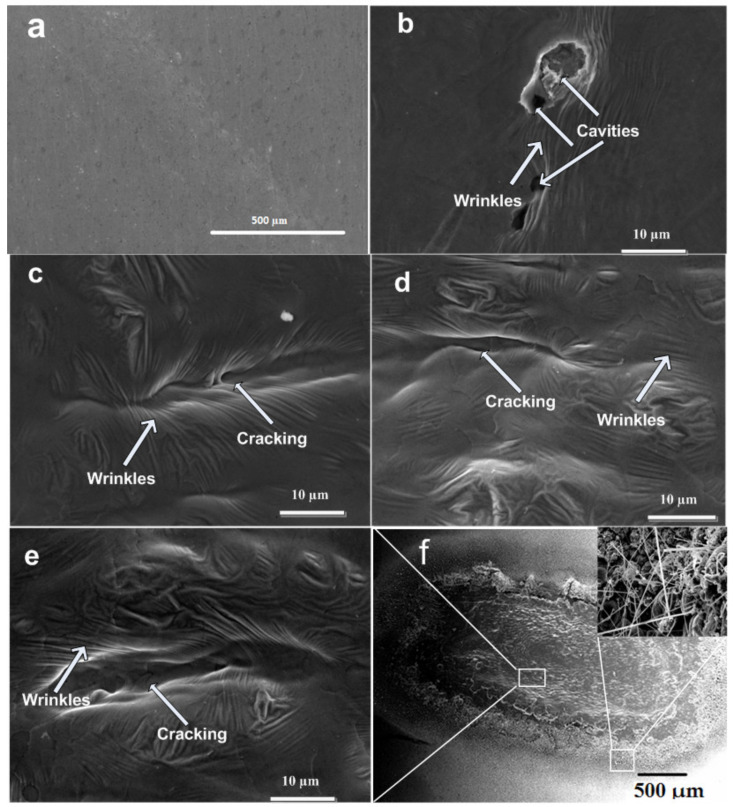
SEM topographical images at center of laser-ablated areas of Cd targets. (**a**) Unirradiated, and irradiated Cd targets for (**b**) 500, (**c**) 1000, (**d**) 1500, (**e**) 2000 laser pulses, and (**f**) full view of ablated region at constant fluence of 3.6 J cm^−2^ in air.

**Figure 3 ijms-23-12749-f003:**
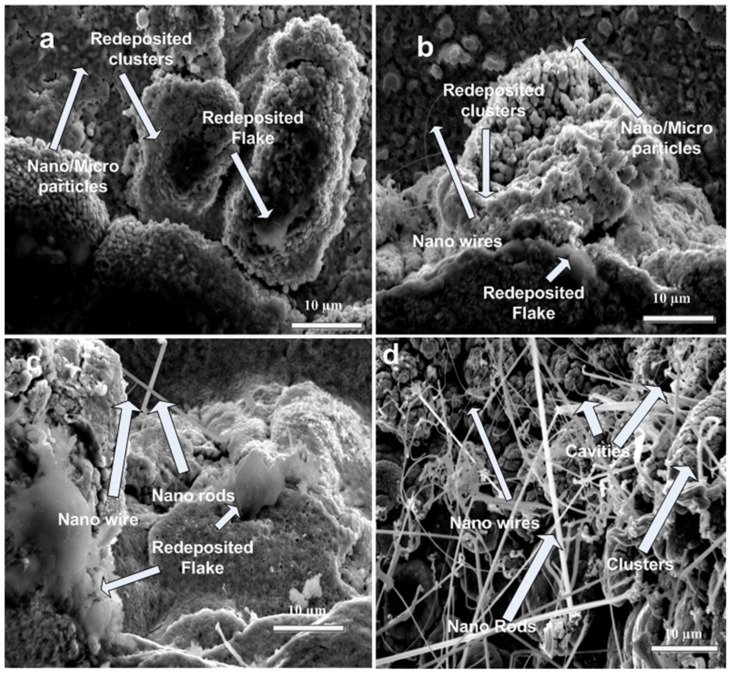
SEM topographical images at periphery of laser-ablated areas of Cd targets for (**a**) 500, (**b**) 1000, (**c**) 1500 and (**d**) 2000 laser pulses, at constant fluence of 3.6 J cm^−2^ in air.

**Figure 4 ijms-23-12749-f004:**
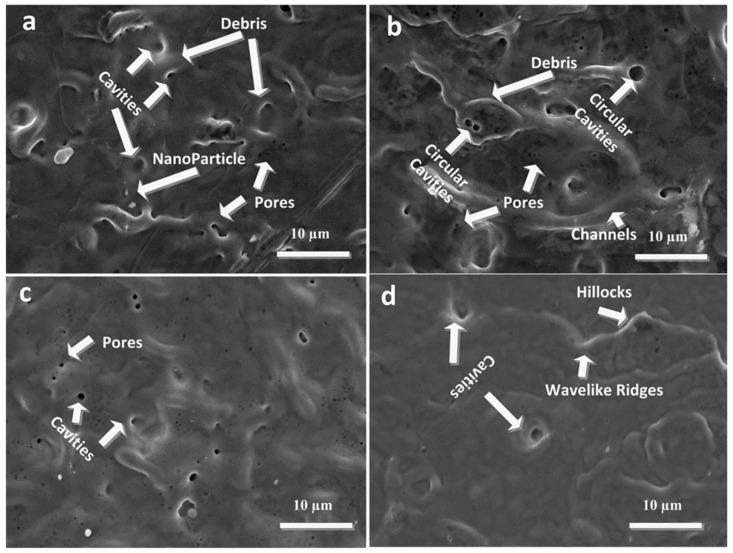
SEM topographical images at center of laser-ablated areas of cd targets using SEM analysis for (**a**) 500, (**b**) 1000, (**c**) 1500 and (**d**) 2000 laser pulses, at constant fluence of 3.6 J cm^−2^ in propanol.

**Figure 5 ijms-23-12749-f005:**
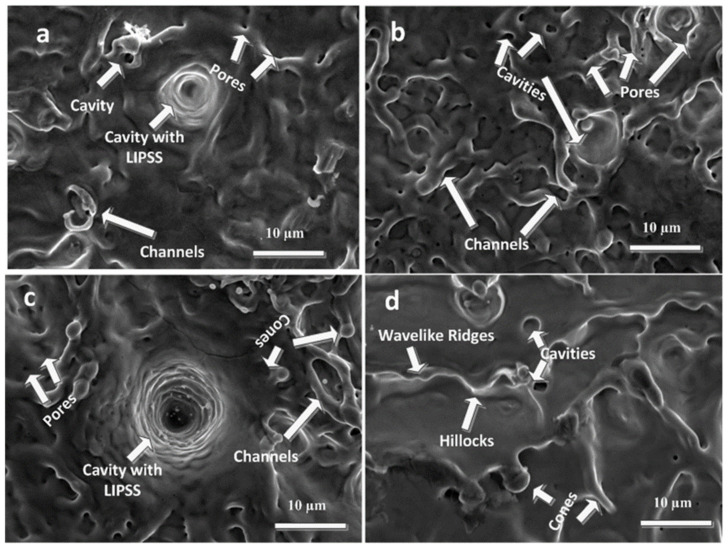
SEM topographical images at periphery of laser-ablated areas of cd targets using SEM analysis for (**a**) 500, (**b**) 1000, (**c**) 1500 and (**d**) 2000 laser pulses, at constant fluence of 3.6 J cm^−2^ in propanol.

**Figure 6 ijms-23-12749-f006:**
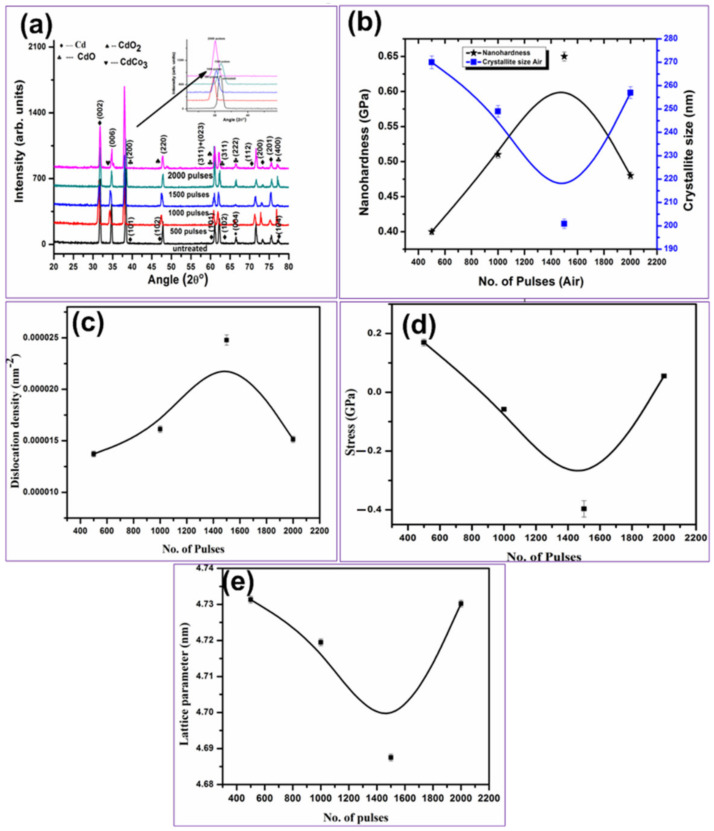
XRD patterns of unirradiated and laser-ablated Cd exposed for 500, 1000, 1500 and 2000 laser pulses, at constant fluence of 3.6 J cm^−2^ in air (**a**) X-ray differactograms, (**b**) the variation in crystallite size and hardness, (**c**) variation in D.D, (**d**) variation in stresses and (**e**) variation in lattice parameters.

**Figure 7 ijms-23-12749-f007:**
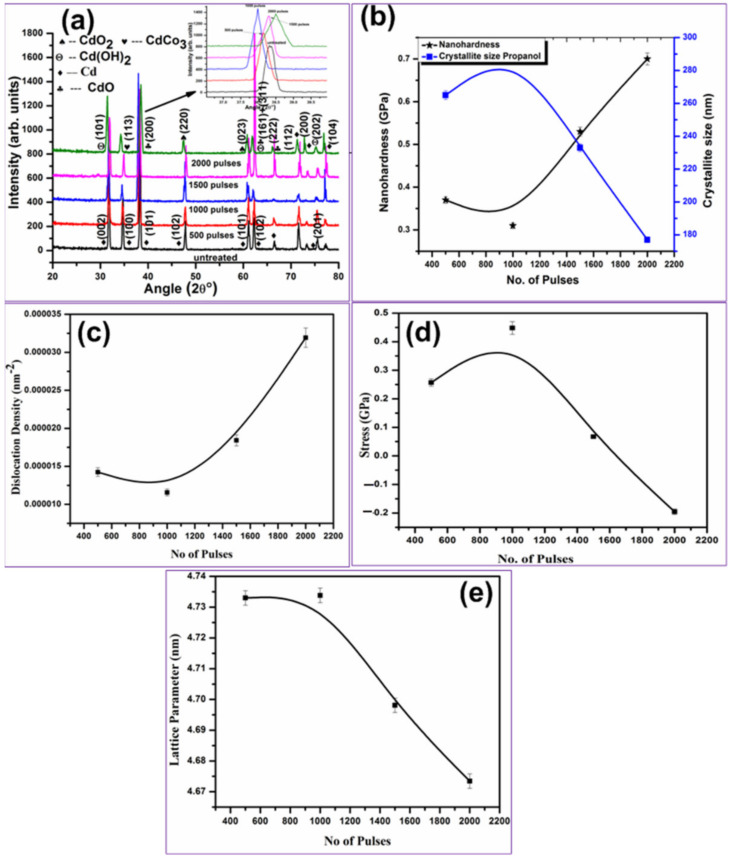
XRD patterns of un-irradiated and laser-ablated Cd exposed for 500, 1000, 1500 and 2000 laser pulses, at constant fluence of 3.6 J cm^−2^ in propanol (**a**) X-ray differactograms, (**b**) the variation in crystallite size and hardness, (**c**) variation in D.D, (**d**) variation in stresses and (**e**) variation in lattice parameters.

**Figure 8 ijms-23-12749-f008:**
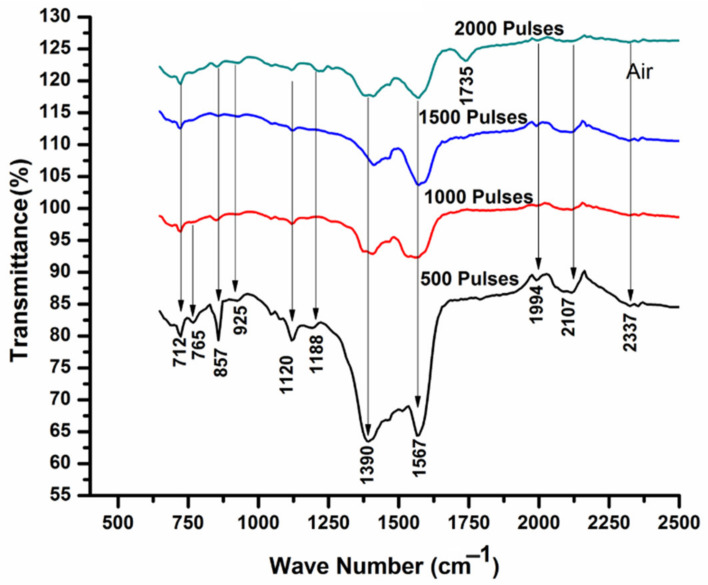
FTIR analysis of laser-ablated Cd exposed for 500, 1000, 1500 and 2000 laser pulses, at constant fluence value of 3.6 J cm^−2^ in air.

**Figure 9 ijms-23-12749-f009:**
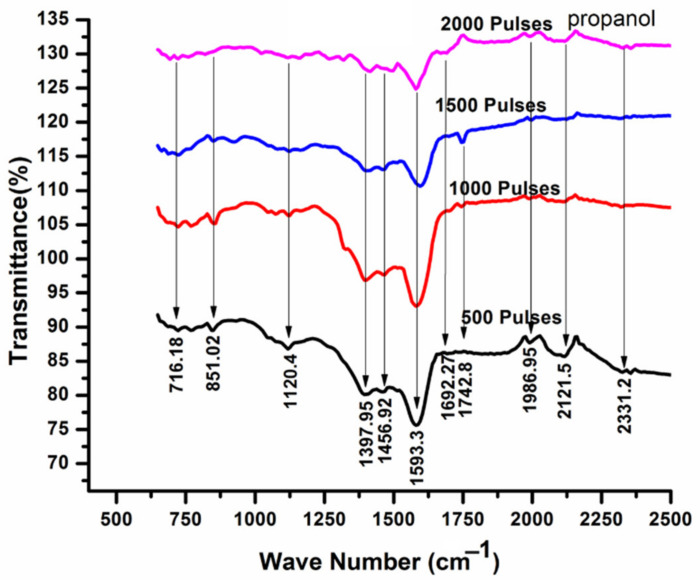
FTIR analysis of laser-ablated Cd exposed for 500, 1000, 1500 and 2000 laser pulses, at constant fluence value of 3.6 J cm^−2^ in propanol.

**Table 1 ijms-23-12749-t001:** The variation in ablated diameter, depth and volume of laser ablated Cd for 500, 1000, 1500 and 2000 laser pulses, at constant fluence of 3.6 J cm^−2^ in air.

Number of Pulses (Air)	Diameter (µm)	Depth (µm)	Volume (10^3^ µm^3^)
500	1118.6	80	23.26
1000	1190	96	29.7
1500	1166.2	90	27.28
2000	1094	144	41.2

**Table 2 ijms-23-12749-t002:** The variation in ablated diameter, depth and ablated volume of laser-ablated Cd for 500, 1000, 1500 and 2000 laser pulses, at constant fluence of 3.6 J cm^−2^ in propanol.

Number of Pulses (Propanol)	Diameter (µm)	Depth (µm)	Volume (10^3^ µm^3^)
500	1237	73	23.47
1000	1309	100	35.73
1500	1190	120	37.43
2000	1047	102	27.76

**Table 3 ijms-23-12749-t003:** The variation in the average peak intensity of IR spectra, at constant fluence value of 3.6 J cm^−2^, in air ambient.

Number of Pulses (Air)	Average Intensity
500	85
1000	94
1500	89.5
2000	88

**Table 4 ijms-23-12749-t004:** Variation in the average peak intensity of IR absorption spectra, at constant fluence value of 3.6 J cm^−2^ in propanol ambient.

Number of Pulses (Propanol)	Average Intensity
500	83
1000	90
1500	87
2000	85

**Table 5 ijms-23-12749-t005:** EDS analysis of untreated and laser-ablated Cd for 500, 1000, 1500 and 2000 laser pulses, at constant fluence value of 3.6 J cm^−2^ in air and propanol environments.

Element Name	Untreated	Air (Weight %)	Propanol (Weight %)
C	1.18	0.45	4.1
N	1.33	1.7	3.28
O	3.34	20.9	15.22
Al	0.96	--	--
Si	0.11	--	--
Cd	90.36	75.45	75.20
W	0.23	--	--
P_a_	2.49	1.5	2.2

## Data Availability

Not applicable.

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
