# Peer review of "Nanosecond Laser Induced Surface Structuring of Cadmium after Ablation in Air and Propanol Ambient"

_ijms, 2022, doi:10.3390/ijms232112749_

Round 1

Reviewer 1 Report

The paper presents observations of changes in the surface of the Cd target as a result of ablation with a laser pulse in air and propanol. There is nothing ground-breaking here. Merely the report of experimental results. However, the results of the experiment may be interesting and useful to some readers.

In the introduction, the authors describe the applications of laser ablation products, while the work itself concerns only changes in the target surface during ablation. So what is the use of this part of the introduction?

Target surface changes have an impact on the ablation process. For example, when a surface is oxidized by the first pulses, the ablation by the next pulse will be of the oxides formed and not the metallic target.

There is necessary extensive editing of English. As a mere single example, the last sentence (page 16, lines 576-578)  is incomprehensible.

The last sentence on page 2 is unclear. “whose value for propanol is calculated and comes out to be 1.5 x 100 10−2/mm for propanol.” – it concerns the absorption coefficient, not a distance.

----------------------------------

It is not clear, why the authors decided to put mathematical formulae into Experimental Setup? Anyway, equations (1, 2, 3) should be corrected and written in a commonly accepted form.

In particular, in Eq. (2):

Lv, should be L_v (lower index v),

tp, should be t_p.  In the formula is t_p, but t (without index) in the explanation text below it

The thermal diffusivity a is first used in Eq. (2) and introduced only later by Eq. (3)

Moreover, formula (2) is valid only when thermal diffusivity is big enough, namely:

                alpha * sqrt(a*t) >> 1

where t – laser pulse duration, a – thermal diffusivity, alpha – absorption coefficient of the target

Author Response

Dear Editor,

We have revised the manuscript according to the reviewer’s comments.

All issues raised are addressed point by point and a file dealing with answers to reviewer’s comments is attached separately.

All changes are incorporated in the manuscript also.

 All the answers to reviewer’s comments are given in RED and all changes made in the manuscript according to reviewer’s comments are also highlighted with RED color.

Regards

Nisar Ali

pen Review

English language and style

(x) Extensive editing of English language and style required
( ) Moderate English changes required
( ) English language and style are fine/minor spell check required
( ) I don't feel qualified to judge about the English language and style

Yes

Can be improved

Must be improved

Not applicable

Does the introduction provide sufficient background and include all relevant references?

(x)

( )

( )

( )

Are all the cited references relevant to the research?

(x)

( )

( )

( )

Is the research design appropriate?

(x)

( )

( )

( )

Are the methods adequately described?

( )

( )

(x)

( )

Are the results clearly presented?

( )

( )

(x)

( )

Are the conclusions supported by the results?

( )

(x)

( )

( )

Comments and Suggestions for Authors

The paper presents observations of changes in the surface of the Cd target as a result of ablation with a laser pulse in air and propanol. There is nothing ground-breaking here. Merely the report of experimental results. However, the results of the experiment may be interesting and useful to some readers.

Q: In the introduction, the authors describe the applications of laser ablation products, while the work itself concerns only changes in the target surface during ablation. So what is the use of this part of the introduction?

Ans: The starting paragraphs in the manuscript are commonly related to the applications and motivations behind the work. In the current work the initial paragraphs are related to importance and applications of pulsed laser ablation, which gives motivation to authors to perform the current experiments. 

The Present work is concerned with laser induced structuring and generation of different by-products (CdO, CdCO3, (Cd(OH)2) after ablation in dry (air) and wet (propanol) environments. PLA in different environments (air and propanol in present case) is the most affective technique for the generation of surface structures, as it doesn’t produce any kind of toxic byproducts. Liquid assisted nanosecond laser ablation of material is more preferable for the fabrication of special morphologies, nanomaterials, phases and microstructures. It is also used for the preparation of single step functionalized phases and nano/micro structures  that are useful in innovative applications in display, optics, and detection fields [1] due to enhanced chemical reactivity, confinement effects and growth of small structures as compared to air. Pulses Laser ablation in liquid (PLAL) is feasible economically, having vide range applications in manufacturing industry and excellent bio-detection properties [2, 3].

Some amendments are also made and highlighted in introduction part of manuscript.

 Q: Target surface changes have an impact on the ablation process. For example, when a surface is oxidized by the first pulses, the ablation by the next pulse will be of the oxides formed and not the metallic target.

Ans: Yes the reviewer is right and authors agree with the reviewer’s point of view that after few pulses the material get oxidized. Even the reactivity of Cd with ambient air is much high that it gets oxidized as it is placed in the ambient air and its surface gets black even after polishing, when we place it in ambient air. Presence of atomic oxygen on untreated surface is also confirmed from EDS analysis.

Q: There is necessary extensive editing of English. As a mere single example, the last sentence (page 16, lines 576-578)  is incomprehensible.

Ans: Changes are being made in the manuscript.

Q: The last sentence on page 2 is unclear. “whose value for propanol is calculated and comes out to be 1.5 x 100 10−2/mm for propanol.” – it concerns the absorption coefficient, not a distance.

Ans: The ambiguity is removed in the manuscript.

----------------------------------

Q: It is not clear, why the authors decided to put mathematical formulae into Experimental Setup? Anyway, equations (1, 2, 3) should be corrected and written in a commonly accepted form.

In particular, in Eq. (2):

Lv, should be L_v (lower index v),

tp, should be t_p.  In the formula is t_p, but t (without index) in the explanation text below it

The thermal diffusivity a is first used in Eq. (2) and introduced only later by Eq. (3)

Ans: The equations are written in commonly accepted form and positions of equations are also corrected as per guideline of reviewer.

Q: Moreover, formula (2) is valid only when thermal diffusivity is big enough, namely:

                alpha * sqrt(a*t) >> 1

where t – laser pulse duration, a – thermal diffusivity, alpha – absorption coefficient of the target

Ans: As the Cd is very reactive in air ambient and get oxidized immediately as we place it in air it is also confirmed from EDS analysis about 3.34 wt% O is present on the untreated target, and also it get more oxidized even after few pulses for both ambient conditions. Due to this reason we took the values of   (16 ´ 105 /cm) [4], and ? (2.925 cm2/sec) for CdO, Here the value of ? is calculated using formula Thermal diffusivity (?) = . Where ? represents the target surface density (for Cd, 8.15 ), C is specific heat (0.3397  for Cd target) and K is thermal conductivity (8.1  for CdO). Where the value of ? comes out to be 2.95 cm2s-1 and the value of tp=20 nsec. So the value of product alpha * sqrt(a*t)  comes out to be 388.6 >> 1, So the that equation 3 is valid for the evaluation of ablation threshold.

Reviewer 2 Report

In this manuscript, the authors studied the laser ablation effect on Cd in air and propanol. Multiple characterizations were performed to analyze the obtained samples. Although this manuscript is well prepared and the pictures are well organized, some serous questions must be answered to get recommendation for publication.

1. This manuscript seems not providing any innovations. The laser ablation effect on metals has already been well studied, both theoretically and experimentally.

2. The PDF card number should be provided in the XRD patterns for better comparison.

3. The conclusion section does not read like conclusions, it’s more like an abstract.

4. Some more recently published works could be cited to enrich the introduction part, such as 10.1016/j.jechem.2021.08.066, 10.1002/aenm.202200855.

Author Response

Dear Editor,

We have revised the manuscript according to the reviewer’s comments.

All issues raised are addressed point by point and a file dealing with answers to reviewer’s comments is attached separately.

All changes are incorporated in the manuscript also.

 All the answers to reviewer’s comments are given in RED and all changes made in the manuscript according to reviewer’s comments are also highlighted with RED color.

Regards

Nisar Ali

Open Review

English language and style

( ) Extensive editing of English language and style required
( ) Moderate English changes required
( ) English language and style are fine/minor spell check required
(x) I don't feel qualified to judge about the English language and style

Yes

Can be improved

Must be improved

Not applicable

Does the introduction provide sufficient background and include all relevant references?

( )

(x)

( )

( )

Are all the cited references relevant to the research?

( )

(x)

( )

( )

Is the research design appropriate?

( )

(x)

( )

( )

Are the methods adequately described?

(x)

( )

( )

( )

Are the results clearly presented?

( )

(x)

( )

( )

Are the conclusions supported by the results?

(x)

( )

( )

( )

Comments and Suggestions for Authors

In this manuscript, the authors studied the laser ablation effect on Cd in air and propanol. Multiple characterizations were performed to analyze the obtained samples. Although this manuscript is well prepared and the pictures are well organized, some serious questions must be answered to get recommendation for publication.

  1. This manuscript seems not providing any innovations. The laser ablation effect on metals has already been well studied, both theoretically and experimentally.

Ans: The aim of present work is to investigate the effect of laser irradiation on surface morphology, structural and mechanical properties of Cd irradiated by KrF Excimer laser. The present work provides us a comparison of the ablation of Cd in air and alcohol (propanol) environments. According to best of our knowledge, laser-induced structuring of Cd in air and propanol environments has not been reported earlier, by any research group, in which surface structuring and enhancement of mechanical properties are discussed in correlation with the changes produced in the crystallinity due to enhancement of chemical reactivity of Cd. In the present work we study how dry and wet environments play a significant role for the surface, structural and mechanical modifications of the material. The wet ablation is always preferable over dry ablation due to confinement effects and enhanced ablation efficiency. The novelty of the present work is that it deals with the growth of various structures and enhancement of nano-hardness in correlation with the changes produced in the crystallinity due to the enhancement of the chemical reactivity of Cd. It provides us a comprehensive insight into the physical processes and mechanisms responsible for defining ablation threshold, surface structuring and enhancement in mechanical properties after ablation in air and propanol. The present study also emphasizes on, how Cd ablation in propanol can modify its chemical composition in more pronounced way as compared to Cd ablation in air. The second aim of this paper is to fabricate carbonates (CdCO3), oxides (CdO) and hydro-oxides (Cd(OH)2) of Cd, which makes it more useful in industrial and biomedical applications after enhancement in its strength, field emission properties and optical absorption. We have observed that propanol assisted ablation is more suitable for the formation of carbonates of Cd (e.g., CdCO3). The mechanical strength of carbonate materials (CdCO3) is significantly higher as compared to simple metallic materials (Cd), making it more useful for industrial applications. By employing pulsed laser ablation in liquids, the synthesis of micro/nanostructured materials becomes possible, which are free from toxic chemicals and by-products, making it useful in biomedical applications.

  1. The PDF card number should be provided in the XRD patterns for better comparison.

Ans: we have used X’Pert High-score data base for XRD phase analysis and have only JCPDS (joint committee on powder diffraction standards) reference numbers that are already inserted in the manuscript.

  1. The conclusion section does not read like conclusions, it’s more like an abstract.

Ans: The modifications are being made in the conclusion section of manuscript.

  1. Some more recently published works could be cited to enrich the introduction part, such as 10.1016/j.jechem.2021.08.066, 10.1002/aenm.202200855.

Ans: The most recently published work (2022) is cited in the manuscript along with the publications mentioned by the reviewer.

Reviewer 3 Report

Comments:

In this work, the authors studied the surface structuring of cadmium in different ambient as air and propanol. The author had well written an article and presented a meaningful discussion of the work. The results and discussion is well written and does not require any changes. As of now, I would highly recommend the publication of this manuscript after a minor revision. I would advise the author to add the schematic diagram for the pulsed laser ablation of Cd. The details are appended below.

1.     The author claimed the experiment 5 cm before the focusing point in the experimental section. Authors can place the schematic diagram or photographic images to interpret the experiment better.

2.     A schematic diagram for the laser-induced transformation of Cd is required.

Author Response

Dear Editor,

We have revised the manuscript according to the reviewer’s comments.

All issues raised are addressed point by point and a file dealing with answers to reviewer’s comments is attached separately.

All changes are incorporated in the manuscript also.

 All the answers to reviewer’s comments are given in RED and all changes made in the manuscript according to reviewer’s comments are also highlighted with RED color.

Regards

Nisar Ali

Open Review

English language and style

( ) Extensive editing of English language and style required
( ) Moderate English changes required
(x) English language and style are fine/minor spell check required
( ) I don't feel qualified to judge about the English language and style

Yes

Can be improved

Must be improved

Not applicable

Does the introduction provide sufficient background and include all relevant references?

(x)

( )

( )

( )

Are all the cited references relevant to the research?

(x)

( )

( )

( )

Is the research design appropriate?

( )

( )

( )

( )

Are the methods adequately described?

(x)

( )

( )

( )

Are the results clearly presented?

(x)

( )

( )

( )

Are the conclusions supported by the results?

(x)

( )

( )

( )

Comments and Suggestions for Authors

Comments:

In this work, the authors studied the surface structuring of cadmium in different ambient as air and propanol. The author had well written an article and presented a meaningful discussion of the work. The results and discussion is well written and does not require any changes. As of now, I would highly recommend the publication of this manuscript after a minor revision. I would advise the author to add the schematic diagram for the pulsed laser ablation of Cd. The details are appended below.

  1. The author claimed the experiment 5 cm before the focusing point in the experimental section. Authors can place the schematic diagram or photographic images to interpret the experiment better.

Ans: The schematic diagram of experimental setup is inserted in the experimental part of manuscript.

  1. A schematic diagram for the laser-induced transformation of Cd is required.

Ans: In Figure 2, untreated Cd surface is also included to show the transformation of Cd after ablation, the full view of ablated target is also inserted in the manuscript.

Reviewer 4 Report

The manuscript presents rather comprehensive, but average-soundness work with some minor drawbacks, which should be corrected prior publication.

First, In Introduction more comparison to PLAL of other materials should be more broadly overviewed, with more impact on nanosecond ablation.

Second, ablation depth and volume measurements versus the number of laser pulses are highly recommended for community merit.

Third, IR absorption band intensities for dry and wet ablation could be better presented in the Table format, as EDX data in Table1. Here, the trace elements (Al, Si, W) missing  after ablation could be deleted at all. The authors should explain/correct the element named Pa.

Author Response

Dear Editor,

We have revised the manuscript according to the reviewer’s comments.

All issues raised are addressed point by point and a file dealing with answers to reviewer’s comments is attached separately.

All changes are incorporated in the manuscript also.

 All the answers to reviewer’s comments are given in RED and all changes made in the manuscript according to reviewer’s comments are also highlighted with RED color.

Regards

Nisar Ali

Open Review

English language and style

( ) Extensive editing of English language and style required
(x) Moderate English changes required
( ) English language and style are fine/minor spell check required
( ) I don't feel qualified to judge about the English language and style

Yes

Can be improved

Must be improved

Not applicable

Does the introduction provide sufficient background and include all relevant references?

( )

(x)

( )

( )

Are all the cited references relevant to the research?

(x)

( )

( )

( )

Is the research design appropriate?

(x)

( )

( )

( )

Are the methods adequately described?

(x)

( )

( )

( )

Are the results clearly presented?

( )

(x)

( )

( )

Are the conclusions supported by the results?

(x)

( )

( )

( )

Comments and Suggestions for Authors

The manuscript presents rather comprehensive, but average-soundness work with some minor drawbacks, which should be corrected prior publication.

Q: First, In Introduction more comparison to PLAL of other materials should be more broadly overviewed, with more impact on nanosecond ablation.

Ans: PLA in different environments (air and liquid in present case) is the most affective technique for the generation of surface structures, as it doesn’t produce any kind of toxic byproducts. Liquid assisted nanosecond laser ablation of material is more preferable for the fabrication of special morphologies, nanomaterials, phases and microstructures. It is also used for the preparation of single step functionalized phases and nano/micro structures  that are useful in innovative applications in display, optics, and detection fields [1] due to enhanced chemical reactivity, confinement effects and growth of small structures as compared to air. Pulses Laser ablation in liquid (PLAL) is feasible economically, for the fabrication of variety of processed materials in manufacturing industry [2, 3].

Various research groups have synthesized different kind of semiconducting materials (like CdO in present work) by using metal targets (like Cd) after irradiation in reactive ambient [5, 6]. Different methods like wet chemical and chemical precipitation methods are employed for the fabrication of these kinds of structures. In reduction step reducing agents are required, for both methods in turn effecting the purity of final product [7] and reduces the suitability of these nano-materials for biomedical applications [8, 9]. Liquid assisted ablation overcomes this issue and causes the generation of micro/nanostructured surfaces that are free from toxic by-products and chemicals [10, 11].

This data is also included in the introduction part of manuscript.

Q: Second, ablation depth and volume measurements versus the number of laser pulses are highly recommended for community merit.

Ans: The values of ablation depth, ablated diameter and ablated volume are calculated and presented in tabular form (table 1 &2) along with discussion, in the manuscript and highlighted with red color.

Q: Third, IR absorption band intensities for dry and wet ablation could be better presented in the Table format, as EDX data in Table1. Here, the trace elements (Al, Si, W) missing after ablation could be deleted at all. The authors should explain/correct the element named Pa.

Ans: The average intensity variation of IR absorption band is also inserted in the form of table (table 3, Table 4) in the manuscript along with discussion.

From EDS analysis (table 4) the elements Al, Si, & W are elements, present on the surface of untreated targets as can be seen from the EDS micrograph. Removal of these elements will disturb the total percentage value i.e., 100 Wt %. So we can’t remove these elements from the tabular data as well.

Round 2

Reviewer 1 Report

The manuscript has been improved considerably. However, it still needs thorough editing. There are some typos and omissions. For example, there is x missing in the exponent of Beer-Lambert formula below figure 1.

Author Response

Dear Editor,

We have revised the manuscript according to the reviewer’s comments.

All issues raised are addressed point by point and a file dealing with answers to reviewer’s comments is attached separately.

All changes are incorporated in the manuscript also.

 All the answers to reviewer’s comments are given in RED and all changes made in the manuscript according to reviewer’s comments are also highlighted with RED color.

Regards

Nisar Ali

Open Review

English language and style

( ) Extensive editing of English language and style required
( ) Moderate English changes required
( ) English language and style are fine/minor spell check required
(x) I don't feel qualified to judge about the English language and style

Yes

Can be improved

Must be improved

Not applicable

Does the introduction provide sufficient background and include all relevant references?

(x)

( )

( )

( )

Are all the cited references relevant to the research?

(x)

( )

( )

( )

Is the research design appropriate?

(x)

( )

( )

( )

Are the methods adequately described?

(x)

( )

( )

( )

Are the results clearly presented?

( )

(x)

( )

( )

Are the conclusions supported by the results?

(x)

( )

( )

( )

Comments and Suggestions for Authors

The manuscript has been improved considerably. However, it still needs thorough editing. There are some typos and omissions. For example, there is x missing in the exponent of Beer-Lambert formula below figure 1.

Ans: All the manuscript is read carefully and removed all mistakes and highlighted in the manuscript as well.

Reviewer 2 Report

This paper has been largely improved, and can be published now.

Author Response

Dear Editor,

We have revised the manuscript according to the reviewer’s comments.

All issues raised are addressed point by point and a file dealing with answers to reviewer’s comments is attached separately.

All changes are incorporated in the manuscript also.

 All the answers to reviewer’s comments are given in RED and all changes made in the manuscript according to reviewer’s comments are also highlighted with RED color.

Regards

Nisar Ali

Yes

Can be improved

Must be improved

Not applicable

Does the introduction provide sufficient background and include all relevant references?

(x)

( )

( )

( )

Are all the cited references relevant to the research?

(x)

( )

( )

( )

Is the research design appropriate?

(x)

( )

( )

( )

Are the methods adequately described?

(x)

( )

( )

( )

Are the results clearly presented?

(x)

( )

( )

( )

Are the conclusions supported by the results?

(x)

( )

( )

( )

Comments and Suggestions for Authors

This paper has been largely improved, and can be published now.

Authors are thankful to the reviewer for such nice decision.